# Facile Synthesis of Functionalized Phenoxy Quinolines: Antibacterial Activities against ESBL Producing *Escherichia coli* and MRSA, Docking Studies, and Structural Features Determination through Computational Approach

**DOI:** 10.3390/molecules27123732

**Published:** 2022-06-10

**Authors:** Mahwish Arshad, Nasir Rasool, Muhammad Usman Qamar, Syed Adnan Ali Shah, Zainul Amiruddin Zakaria

**Affiliations:** 1Department of Chemistry, Government College, University Faisalabad, Faisalabad 38000, Pakistan; mahwishnadeem001@gmail.com; 2Department of Microbiology, Government College University, Faisalabad 38000, Pakistan; musmanqamar@gcuf.edu.pk; 3Faculty of Pharmacy, Universiti Teknologi MARA Cawangan Selangor Kampus Puncak Alam, 42300 Bandar Puncak Alam, Selangor, Malaysia; syedadnan@uitm.edu.my; 4Atta-ur-Rahman Institute for Natural Product Discovery (AuRIns), Universiti Teknologi MARA Cawangan Selangor Kampus Puncak Alam, 42300 Bandar Puncak Alam, Selangor, Malaysia; 5Department of Biomedical Sciences, Faculty of Medicine and Health Sciences, Universiti Malaysia Sabah, Jalan UMS, Kota Kinabalu 88400, Sabah, Malaysia

**Keywords:** 4-methyl phenylboronic acid, Chan–Lam coupling, antibacterial activity, molecular docking, NLO properties, ^1^H NMR comparison, ESBL, MRSA

## Abstract

The synthesis of new 6-Bromoquinolin-4-ol derivatives (**3a**–**3h**) by Chan–Lam coupling utilizing different types of solvents (protic, aprotic, and mixed solvents) and bases was studied in the present manuscript. Furthermore, their potential against ESBL producing *Escherichia coli* (ESBL *E. coli*) and methicillin-resistant *Staphylococcus*
*aureus* (MRSA) were investigated. Commercially available 6-bromoquinolin-4-ol (**3a**) was reacted with different types of aryl boronic acids along with Cu(OAc)_2_ via Chan–Lam coupling methodology utilizing the protic and aprotic and mixed solvents. The molecules (**3a**–**3h**) exhibited very good yields with methanol, moderate yields with DMF, and low yields with ethanol solvents, while the mixed solvent CH_3_OH/H_2_O (8:1) gave more excellent results as compared to the other solvents. The in vitro antiseptic values against ESBL *E. coli* and MRSA were calculated at five different deliberations (**10**, **20**, **30**, **40**, **50** mg/well) by agar well diffusion method. The molecule **3e** depicted highest antibacterial activity while compounds **3b** and **3d** showed low antibacterial activity. Additionally, MIC and MBC standards were calculated against the established bacteria by broth dilution method. Furthermore, a molecular docking investigation of the derivatives (**3a**–**3h**) were performed. Compound (**3e**) was highly active and depicted the least binding energy of −5.4. Moreover, to investigate the essential structural and physical properties, the density functional theory (DFT) findings of the synthesized molecules were accomplished by using the basic set PBE0-D3BJ/def2-TZVP/SMD water level of the theory. The synthesized compounds showed an energy gap from 4.93 to 5.07 eV.

## 1. Introduction

The emergence and spread of multidrug-resistant bacteria is a global public health concern due to limited available therapeutic options. Extended spectrum-β-lactamase producing (ESBL *E. coli*), and methicillin-resistant *Staphylococcus aureus* (MRSA) produce resistance to β-lactam inhibitors and β-lactamasequinoline, due to the acquisition of various antimicrobial resistant genes, leaving limited available treatment options. These pathogens cause serious clinical toxicities and contagions such as bacteremia, septicemia, convoluted contagions, urinary tract infections, skin, and soft tissue infections. It was estimated that by 2050, the death rate will increase to >100 million and the economic loss will be >100 trillion $, if the antimicrobial-resistant bacteria persist. The World Health Organization ranked ESBL and MRSA on the high priority pathogens list. Therefore, therapeutic approaches are needed to tackle the AMR problem [1].

Quinoline is a heterocyclic compound found abundantly in numerous nature-made substances such as cinchona alkaloids as well as in pharmacological dynamics, exhibiting a comprehensive collection of biological activity. Chloroquine, primaquine, mefloquine, and quinine constitute notable examples of the drugs containing the quinoline scaffold that are used for the treatment of malaria [2,3]. Floroquinolines, which are closely related to the quinoline ring, constitute another important class of antimicrobial agents used for the treatment of the infections caused by especially Gram-negative bacteria [4,5]. Fluoroquinolones have been recommended and used for a variety of stable or decreasing incidence throughout most reinfections including those of the respiratory, gastro-regions in the world and an increasing incidence in intestinal (GI) and genitourinary tracts, and must have anti-TB activity [6,7,8]. Mefloquine is a well-recognized compound with quinoline structure and is extensively recommended for the prophylaxis of chloroquine-resistant *Plasmodium falciparum* malaria [9]. Quinolines also have credible coordinating ability and virtuous metal recognition properties [10]. As a result, their derivatives depict a wide spectrum of applications in the areas of drug discovery [11], materials science [12,13], and catalysis [14,15].

A study published by Rbaa and coworkers described the synthesis of new 4-aryl-3,4-dihydro2H-[1,3]oxazino[5,6-h]quinolin-2-one, which gave remarkable antibacterial activity against *E. chloacae*, *K. pneumoniae*, *S. aureus*, *A. baumanii*, and *E. coli* [16]. Ramesh et al. synthesized novel derivatives of tetrahydro quinoline annulated heterocycles, which showed remarkable antibacterial activities against *Staphylococcus aureus, E. coli, Pseudomonas aeruginosa, Klebsiella pneumonia, Salmonella typhimurium*, and *Bacillus subtilis* [17]. The aim of our study was to synthesize the functionalized quinolines which would have great significance against the WHO (World Health Organization) hit list of pathogens, i.e., Acinebacter, Pseudomonas, and various Enterobacteriaceae (including (*Klebsiella, E. coli, Serratia*, and *Proteus*). In light of the therapeutic significance of quinolines, we synthesized a new series of the functionalized derivatives of 6-bromoquinolin-4-ol (**1a**) by O-alkylated ether synthesis by the Chan–Lam C–O cross-coupling approach at room temperature using copper salt and aryl boronic acids. For the synthesis of new compounds (**3a**–**3h**) by Chan–Lam coupling, we utilized different types of solvents, i.e., protic, aprotic, and mixed solvents, and the activity of the catalyst was also studied by replacing the Cu(OAc)_2_ with Cu(OAc)_2_.H_2_O (copper (II) acetate monohydrate). Moreover, the antibacterial activities of the synthesized derivatives were analyzed against ESBL producing *E. coli* and MRSA. MIC and MBC results were estimated and additionally, the molecular docking study was executed to determine their pharmaceutical potential. Moreover, to investigate their electronic and structural properties, density functional theory (DFT) studies were performed. Herein, our studies related to this scientific approach have not been reported so far.

## 2. Results and Discussion

### 2.1. Chemistry

Initially, we performed the preliminary reaction with 6-bromoquinolin-4-ol (**1**) (1 eq.), 4-methyl phenylboronic acid (**2**) (1 eq.), Et_3_N (1 mol%), 1 eq. of Cu(OAc)_2_ and molecular sieves of 4 Å in the aerobic atmosphere, stirring for 48 h at room temperature (Figure 1). We optimized the concentration of boronic acid and the catalyst Cu(OAc)_2_ from 1 eq. to 3 eq. It was noteworthy that Cu(OAc)_2_ 2 eq. gave the optimum results. The purification of the reaction mixture by running the mixture in silica gel (40:60 of hexane:EtOAc) column was done to obtain the anticipated invention **3a** in 89% yield. The structural representation of final compounds (**3a**–**3h**) i.e the derivatives of 6-bromoquinolin-4-ol by Suzuki coupling reaction is provided in Figure 1 and the general scheme for the arylation of of 6-bromoquinolin-4-ol is represented in Figure 1.

Additionally, we observed that solvent plays a crucial role to enhance the reactivity between the reacting species. Therefore, we optimized the protic and the aprotic solvents by utilizing Et_3_N, TMEDA, and pyridine as a base. During the reaction, when we utilized Et_3_Nbase with methanol solvent (the protic solvent), we revealed good to very good yields (57–92%). Nevertheless, we observed that good results were obtained due to better solubility and hence better interaction between methanol and the reacting species. Surprisingly, a low (31–61%) yield was obtained in ethanol, while in DMF (the aprotic solvent), a moderate (51–73%) yield was obtained. This trend can also be attributed to better solubility in DMF (Table 1). Furthermore, when TMEDA as a base with methanol was used, we observed the same trend to provide very good yields, as with Et_3_N. Methanol gave the best results (59–73%); however, ethanol showed moderate yields (31–57%) as shown in Table 1. Therefore, we can state that methanol is the best solvent and Et_3_N base for this type of coupling reaction. Herein, it was observed that pyridine did not show any considerable results.

In addition, we noticed the activity of the catalyst. When Cu(OAc)_2_ was replaced by Cu(OAc)_2_.H_2_O (copper (II) acetate monohydrate), the yield of the coupling product was dramatically enhanced to 97–79% without adding the 4 Å molecular sieves in the reaction mixture. The reaction time was reduced to 45 min as compared to a long time duration, i.e., 48 h [18]. Nevertheless, water has gained a lot of attention as a solvent due to its pseudo-organic nature and this is good for the solvation of the base, with very good results [19,20]. In addition, we checked the CH_3_OH/H_2_O mixed solvent ratio by using Et_3_N and TMEDA as a base (Table 2). Herein, the ratio of the mixed solvent CH_3_OH/H_2_O was optimized by using 8:1, 6:1, 4:1, and 2:1, respectively. Notably, the mixed solvent 8:1 showed very good results with TMEDA and Et_3_N bases (Table 2).

All quinoline derivatives (**3a**–**3h**) were obtained in good to excellent yields. All these derivatives were confirmed by IR and NMR. The IR spectra of all derivatives exhibited the related functional groups peaks. The ^1^H-NMR spectrum of **3a** showed four doublet peaks at δ 8.43, 8.14, 8.05, and 7.0 ppm and two doublet of doublet peaks at δ 7.67 and 6.81 ppm due to aryl protons. The methyl protons attached at 4 position of the phenyl ring exhibited a singlet peak of 3H at δ 2.34 ppm. The ^13^C NMR spectrum showed all related peaks along with a peak at δ 160.74 ppm due to a carbon attached with ether group and a peak at δ 21.12 due to methyl carbon. The proton spectrum of **3b** showed four doublet peaks at δ 8.44, 8.14, 8.05, and 6.73 ppm, one doublet of doublet peaks at δ 7.67 ppm, and one multiplet in the 6.86–6.79 ppm region due to aryl protons. The methoxy protons exhibited a singlet peak of 3H at δ 3.81 ppm. The ^13^C NMR spectrum also showed a peak at 56.03 due to methoxy carbon. The proton spectrum of **3c** revealed four doublet peaks at δ 8.68, 8.41, 7.81, and 6.89 ppm, two doublet of doublet peaks at δ 8.57 and 7.07 ppm, and two multiplets in the 8.05–8.0 and 7.54–7.45 ppm regions due to aryl protons. The proton spectrum of **3d** showed three doublet peaks at δ 8.21, 8.07, and 8.02 ppm, three doublet of doublet peaks at δ 7.65, 7.49, and 7.35 ppm, and two multiplets in 7.98–7.92 and 7.17–7.12 ppm region due to aryl protons. The **3e** proton spectrum exhibited three doublet peaks at δ 8.45, 8.12, and 8.05 ppm, two doublet of doublet peaks at δ 7.67 and 6.82 ppm due to aryl protons, and one triplet peak at 6.98 ppm due to phenyl proton at 4 position. The proton spectrum of **3f** showed three doublet peaks at δ 8.45, 8.13, and 8.05 ppm, two doublet of doublet peaks at δ 7.67 and 6.92 ppm, one triplet at 6.82 ppm, and one doublet of doublet of doublet at 6.70 ppm due to aryl protons. The proton spectrum of **3g** showed six doublet peaks at δ 8.47, 8.13, 8.05, 7.06, 6.86, and 6.74 ppm, and one doublet of doublet peaks at δ 7.67 ppm due to aryl protons. In addition, the thio-methyl protons attached at 4 position of phenyl ring exhibited a singlet peak at 2.63 ppm. The ^13^C NMR spectrum showed a peak at 16.58 due to thio-methyl carbon. The proton spectrum of **3h** showed six doublet peaks at δ 8.47, 8.14, 8.05, 7.81, 6.96, and 6.85 ppm, and one doublet of doublet peak at δ 7.67 ppm due to aryl protons. The ester protons attached at 4 position of phenyl ring exhibited a singlet peak at 3.95 ppm. Its ^13^C NMR spectrum showed confirmation peak at 167.35 ppm due to carbonyl carbon and a peak at 52.08 is due to methoxy carbon.

### 2.2. Antibacterial Activity of the Compounds against XDR Pathogens

The molecules **3a**–**3h** were examined for antibacterial activity at five different deliberations (10, 20, 30, 40, and 50 mg/well) against ESBL producing *E. coli* and MRSA by agar well diffusion method. The broth dilution method was practiced documenting the MIC and MBC values. The effects of the dilutions are unveiled in Table 3a,b and Figure 2 and Figure 3 concerning the activity revealed that both bacteria are liable in contradiction of the established molecules. The concentration of the compounds increases proportionally as the zone of inhibition (mm) increases. Compound **3e** revealed the highest region of inhibition (21 and 20 mm) at 50 mg concentration as compared to other compounds against both bacteria while cefoxitin disc antibiotic was inhibited at a 4 mm zone of inhibition. The compounds **3b** and **3d** presented low activity in comparison to the other compounds. Conversely, the other molecules depicted better commotion. The values of MIC and MBC against both bacteria and results indicate that compound **3e** exhibited MIC and MBC of 6.25 and 12.5 mg respectively against ESBL producing *E. coli* and 3.125 and 6.5 mg respectively against MRSA while all other compounds also showed remarkable value (Table 4a,b, Figure 4 and Appendix A).

#### Molecular Docking Results

Molecular docking studies were performed to validate the biological activity of the synthesized compounds against X-ray crystallographic structure of resistant *Escherichia coli* strain (PDB: 2Y2T) *E. coli* produce functional amyloid fibers termed as curli, which enable the biofilm formation, mediate cell–cell and cell surface interactions, and enhance the pathogenicity [18]. This emphasizes the importance of investigating the structural and biochemical properties of curli when formed natively by major fiber subunit protein, CsgA [21]. Moreover, 2Y2T is the only *E. coli* CsgC structure in reduced form available in PDB. Thus, we examined the interactions of 2Y2T with the phenoxy quinolone derivatives to identify the interactions associated with the biological activity of highly active compounds. The docking simulation of highly active (**3e**) shows that the pyridine moiety represents hydrogen bonding (HN—OH) with Lys78 and hydrophobic (pi—H) interaction with Val80. Additionally, the 2-chlorobenzene in **3e** also showed the tendency of hydrogen bonding with the Val77. Interestingly, a similar interaction pattern was observed for moderately active **3h** except for the hydrogen bonding with Val77. However, the pyridine moiety of **3c** (moderately active) shows hydrogen bonding with Thr26 (NH—OH) and Cys29 (CH—SG). The **3b** (least active) compounds only showed hydrogen bonding (HN—OH) between and hydrophobic interactions with the nitrogen of the pyridine ring and Lys78 residue. Moreover, **3d** (least active) formed a hydrophobic (pi-H) interaction between the pyrimidine moiety and Trp95 residue. Further to this, the least binding energy (−5.4) was observed for highly active (**3e**) in comparison to the other compounds of the series (Table 5, Figure 5).

From the above docking analysis, it can be concluded that hydrogen bonding and hydrophobic interaction with the Lys78, Val77, and Val80 are implicated as important for the improved biological activity of the **3e** compound. In addition, the least binding energy was also observed for the **3e** compound. It is evident from the docking analysis that the in silico studies are also in agreement with in vitro studies.

### 2.3. Computational Studies

DFT studies have been proven to be a powerful tool to calculate the different electronic characteristics of compounds, which is why all the synthesized compounds were analyzed structurally by the density functional theory (DFT) study. All the 6-bromoquinolin-4-ol byproducts **3a**–**3h** with different functional groups by reacting with varied boronic acids were scrutinized in 3D and optimized by DFT calculations, utilizing the basic set PBE0-D3BJ/def2-TZVP/SMD water level of the theory [22]. A hybrid density functional (PBE0) with 25% Hartree Fock exchange was used Triple ζ basis set Def2-TZVP by experiential dispersal alteration (D3) by Grimme. The solvent effect was investigated through the polarizable continuum model (PCM). Model density (SMD) parameter was set by Truhlar. In all computations, water was used as a solvent. No fictional frequencies established the structures to be factual minima. Gauss View 6 software was used to determine the structural conceptions.

#### 2.3.1. Optimized Geometries

Geometries optimization plays a vital role to predict the geometries of the compounds in the ground state and it proves very helpful to calculate the other properties of the compounds. A conformational examination is a fundamental methodology to predict the lowest potency conformers, which is essential to figure out all the other properties of the conformational flexible compounds. A hassle-free probable energy examination was performed by examining the significant dihedrals, and the derivatives with very low values of energy were nominated from the consequential potential energy scan (PES), after which the geometry optimization and elimination of replicas were carried out. Figure 6 provides the improved geometries of the most constant conformers of the molecules (**3a**–**3h**) deliberated at the PBE0-D3BJ/def2-TZVP/SMD water level of theory.

#### 2.3.2. NMR Spectra

NMR spectroscopy is a spectroscopic technique employed to govern the physical and chemical properties of the atomic nuclei. This technique provides comprehensive statistics about the electronic structure of the molecule and its functional groups, dynamics, reaction state, resonance frequency, and chemical environment [23].

In this research work, the investigational ^1^H ranges (**3a**–**3h**) were documented at the PBE0-D3BJ/def2-TZVP/SMD water level of theory, whereas the evaluation of the ^1^H and ^13^C NMR ranges was accomplished by utilizing the same optimization methodology. The assessment of the tentative and hypothetical values of ^1^H NMR data of the derivative **3a** is depicted in Table 6 while the ^1^H NMR records of compounds **3b**–**3h** are given in the Appendix A. It is evident from Table 6 that the NMR analysis was precise, and the mean absolute error (MAE) is only 0.18 ppm.

#### 2.3.3. Frontier Molecular Orbital (FMO) Analysis and Hyperpolarizability

The energies of FMOs are an indispensable part of DFT calculation used to elucidate the chemical stability and the electronic and optical possessions of molecules [24,25]. In conjugated systems, the lowest unoccupied molecular orbital (LUMO) and the highest occupied molecular orbital (HOMO) define plentiful interfaces among molecules, categories of reaction, the UV–visible spectrum, and fluorescence [26,27,28]. From the value of E_HOMO_, we can predict the nucleophilic nature and by the value of E_LUMO,_ the electrophilic nature of the compound can be imagined [29]. The HOMO energy (E_HOMO_), LUMO energy (E_LUMO_), and HOMO–LUMO energy gap (E_LUMO_—E_HOMO_) are the substantial aspects for the prediction of general reactivity of the molecule [30], chemical softness, rigidity, constancy, and physical appearance of the elements and compounds under study. Materials with more HOMO–LUMO energy difference (E_gap_) are known as rigid and stiff molecules with elevated kinetic strength and less chemical reactivity [31]. On the other hand, compounds with less HOMO–LUMO energy difference are even and flexible molecules with lesser stability and greater reactivity [32].

The compounds **3a**–**3h** have nearly a similar electronic configuration and pattern but some are a little bit different. All the compounds have the probability of the electrons spread over the heterocyclic rings, but in some compounds, the probability of finding the electrons is extended above the phenyl ring. A scheme representing these shells is shown in Figure 7.

In this series (**3a**–**3h**), it was observed that compounds **3a**, **3e**, **3f**, and **3h** consisting of heterocyclic rings with electron-withdrawing groups possess a HOMO–LUMO energy gap of 5.07 eV, which is the high value among other derivatives, and consequently these compounds depict very low reactivity. The derivative **3g** with the aromatic cyclic rings with the electron-donating group has the lowest HOMO–LUMO energy difference of 4.93 eV, and hence it is the most reactive among the series. The remaining compounds have a HOMO–LUMO energy difference in the range of 4.95–4.99 eV. The values of HOMO and LUMO energies and their energy difference as well as the hyper-polarizability (β) of all the compounds **3a**–**3h** are documented in Table 7.

#### 2.3.4. Molecular Electrostatic Potential

To predict the physical appearance, magnitude, and dipole moment of the compounds, molecular electrostatic potential (MESP) maps play a crucial role. As seen in Figure 8, the positions with a high probability of electrons are displayed in red, whereas the blue color demonstrates the low-electron density positions [33]. Figure 8 is a representation of the MESP subversions of compounds **3a**–**3h**, the bulkiness and magnitude of the molecule and its electron-high as well as its electron-deficient positions can be effortlessly imagined, which may help in designing reactions of the molecules under study.

It is evident from the plots of the MESP of compounds **3a**–**3h** that the red color shows that the electron-donating site is predominantly positioned above the quinoline group, which is the most negative ESP site and hence a feasible electrophilic attacking position, while the blue colors are principally positioned above the boronic acid of the phenyl group which is the most positive ESP section and hence feasible for the electrophilic attack.

#### 2.3.5. Global Reactivity Parameters

The energy figures and values of FMOs (*E*_gap_ = *E*_LUMO_ − *E*_HOMO_) are worthwhile for the prediction of global reactivity information, including global hardness (η), global softness (σ), global electrophilicity index (ω), electron affinity (EA), ionization potential (IP), electronegativity (X), and the chemical potential (µ) [34,35,36,37]. IP and EA can be calculated by utilizing Equations (1) and (2).
IP = −E_HOMO_(1)
EA = −E_LUMO_(2)

The chemical hardness (η), chemical potential (µ), and electronegativity (X) are investigated by employing Koopmans’s theorem and the following Equations (3)–(5) [38].
X = [IP + EA] = −[E_LUMO_ + E_HUMO_](3)
η = [IP − EA]/2 = −[E_LUMO_ − E_HUMO_]/2(4)
µ = [E_LUMO_ + E_HUMO_]/2(5)

The global softness (σ) is described by Equation (6). An electrophilicity index (ω) was introduced by Parr et al., as in Equation (7) [39].
σ = 1/2η(6)
ω = µ2/2η(7)

The lowest figures of ionization energy (6.47 eV) and electron affinity (1.52 eV) of the compound **3b** depicted in Table 8 certifies their extraordinary reactivity and unstable nature in the series **3a**–**3h**.

From the series of all derivatives, **3g** has the assessment of η 2.46 eV which is the lowest rate, and the high rate of σ (0.20 eV). Therefore, it is chemically lenient and flexible (most reacting species). However, **3a, 3e, 3f,** and **3h** have a peak value of η 2.53 eV and a lowermost value of σ 0.19 eV, hence they are less reactive. These outcomes are associated with the HOMO–LUMO energy difference of all the manufactured compounds.

Compound **3b** has a maximum electronic chemical prospective value of −4.00 eV, while **3e** has the lowest chemical prospective value −4.12 eV. The outcomes showed that **3b** has the lowest electrophilic nature and showed the value of 3.23 eV, whereas **3c** and **3e** have the highest values of 3.34 eV and possess an intensely electron-loving capacity.

## 3. Materials and Methods

### 3.1. General Information

All the chemicals and elements used in all experiments were of analytical grade and were purchased from Shangai Macklin Biochemical Co., Ltd. (Shangai, China) and Alpha Aeser by Thermo Fisher Scientific (Waltham, MA, USA). The optimum reaction condition for the Chan–Lam coupling in an inert atmosphere and this reaction condition was maintained to obtain a high percentage of the products. The melting points of the synthesized products were obtained by using a Buchi-B-540 melting point apparatus (New Castle, DE, USA), and IR spectras were obtained by using AIM 9000, FTIR spectrophotometer. The completion of the reactions was examined by thin-layer chromatography using silica gel plates and monitored by 254 nm UV light. The ^1^H NMR spectra were recorded on a Bruker Avance III 600 Ascend spectrometer using BBO probe, while ^13^C NMR spectra were recorded on Bruker Avance III 600 Ascend spectrometer operating at 150 MHz using DMSO-d_6_ solvent. For elemental analysis, CHN Perkin-Elmer 2400 series analyzer was used.

### 3.2. General Method for the Coupling of 6-Bromoquinolin-4-ol with Aryl Boronic Acid

In a Schlenk tube, the three compounds, i.e., 6-bromoquinolin-4-ol (1 equivalent), aryl boronic acid (2 eq.), and Cu(OAc)_2_ (1 eq.), were taken along with the 10 mL of solvent, and it was stirred in an exposed and aerobic atmosphere for 20 min at room temperature. After 20 min, Et_3_N base (2 eq.) was also inoculated in the Schlenk tube through a syringe, the reaction was left to be stirred for 48 h. After completion, the workup was done; i.e., first of all, the reaction mixture was filtered to eradicate the impurities and to obtain the maximum possible product, then the reaction mixture was concentrated by evaporating the extra solvent by maintaining high temperature and low pressure at the rotary evaporator and finally the product was obtained in the dry and crystalline form. The column chromatography using n-hexane and ethyl acetate in a 40:60 ratio was run to purify the product. Finally, the production of **3d** was confirmed by the structure elucidation via ^1^H NMR, ^13^C NMR, IR, and elemental analysis [40,41,42].

### 3.3. Characterization Data

6-bromo-4-(p-tolyloxy)quinoline (**3a**). M.p.: 267–269 °C. IR (KBr, cm^−1^): 2952, 2921, 2865, 1457, 1096, 945. ^1^H NMR (600 MHz, DMSO-d_6_) δ: 8.43 (d, *J* = 7.5 Hz, 1H), 8.14 (d, *J* = 1.6 Hz, 1H), 8.05 (d, *J* = 7.5 Hz, 1H), 7.67 (dd, *J* = 7.5, 1.4 Hz, 1H), 7.00 (d, *J* = 7.5 Hz, 2H), 6.81 (dd, *J* = 19.6, 7.5 Hz, 3H), 2.34 (s, 3H). ^13^C NMR (150 MHz, DMSO-d_6_) δ: 160.74, 153.34, 151.07, 145.40, 134.39, 134.17, 129.72, 129.42, 127.23, 118.71, 117.41, 115.27, 107.65, 21.12. Anal. Calcd for the C_16_H_12_BrNO: C, 61.17; H, 3.85; N, 4.46; O, 5.09. Found: C, 61.11; H, 3.89; N, 4.50; O, 5.12.

6-bromo-4-(4-methoxyphenoxy)quinoline (**3b**). M.p.: 282–283 °C. IR (KBr, cm^−1^): 3270, 2955, 2923, 1634, 1618, 1070. ^1^H NMR (600 MHz, DMSO-d_6_) δ: 8.44 (d, *J* = 7.5 Hz, 1H), 8.14 (d, *J* = 1.4 Hz, 1H), 8.05 (d, *J* = 7.5 Hz, 1H), 7.67 (dd, *J* = 7.5, 1.4 Hz, 1H), 6.86−6.79 (m, 3H), 6.73 (d, *J* = 7.5 Hz, 2H), 3.81 (s, 3H). ^13^C NMR (150 MHz, DMSO-d_6_) δ: 160.74, 156.51, 151.07, 150.18, 145.40, 134.17, 129.72, 127.23, 120.03, 117.41, 115.27, 114.85, 107.65, 56.03. Anal. Calcd for the C_16_H_12_BrNO_2_: C, 58.20; H, 3.66; N, 4.24; O, 9.69. Found: C, 58.14; H, 3.69; N, 4.29; O, 9.73.

6-bromo-4-(pyridin-2-yloxy)quinoline (**3c**). M.p.: 308–310 °C. IR (KBr, cm^−1^): 3210, 2260, 1653, 1596, 1309, 1187, 1149, 1097, 705. ^1^H NMR (600 MHz, DMSO-d_6_) δ: 8.68 (d, *J* = 6.4 Hz, 1H), 8.57 (dd, *J* = 7.5, 1.3 Hz, 1H), 8.41 (d, *J* = 7.5 Hz, 1H), 8.05−8.0 (m, 1H), 7.81 (d, *J* = 1.4 Hz, 1H), 7.54−7.45 (m, 1H), 7.07 (dd, *J* = 12.0, 4.5 Hz, 2H), 6.89 (d, *J* = 7.5 Hz, 1H). ^13^C NMR (150 MHz, DMSO-d_6_) δ: 161.07, 153.30, 150.05, 148.71, 144.34, 139.51, 135.87, 130.72, 128.45, 122.68, 118.15, 117.74, 113.67, 108.38. Anal. Calcd for the C_14_H_9_BrN_2_O: C, 55.84; H, 3.01; N, 9.30; O, 5.31. Found: C, 55.78; H, 3.05; N, 9.32; O, 5.33.

6-bromo-4-(thiophen-3-yloxy)quinoline (**3d**). M.p.: 310–311 °C. IR (KBr, cm^−1^): 2953, 2925, 2871, 1654, 1595, 1507, 1277, 1147, 1090, 830, 769. ^1^H NMR (600 MHz, DMSO-d_6_) δ: 8.21 (d, *J* = 7.5 Hz, 1H), 8.07 (d, *J* = 1.2 Hz, 1H), 8.02 (d, *J* = 6.7 Hz, 1H), 7.98−7.92 (m, 1H), 7.65 (dd, *J* = 8.5, 1.6 Hz, 1H), 7.49 (dd, *J* = 7.2, 1.2 Hz, 1H), 7.35 (dd, *J* = 7.5, 2.9 Hz, 1H), 7.17 −7.12 (m, 1H). ^13^C NMR (150 MHz, DMSO-d_6_) δ: 163.09, 153.67, 150.08, 143.56, 137.57, 134.09, 130.23, 127.40, 123.99, 118.99, 117.41, 110.67, 106.81. Anal. Calcd for the C_13_H_8_BrNOS: C, 51.00; H, 2.63; N, 4.57; O, 5.23. Found: C, 50.93; H, 2.66; N, 4.59; O, 5.25.

6-bromo-4-(3,5-dichlorophenoxy)quinoline (**3e**). M.p.: 325–327 °C. IR (KBr, cm^−1^): 3095, 2954, 2923, 2853, 1458, 1261, 1235, 1065, 811. ^1^H NMR (600 MHz, DMSO-d_6_) δ: 8.45 (d, *J* = 7.5 Hz, 1H), 8.12 (d, *J* = 1.4 Hz, 1H), 8.05 (d, *J* = 7.5 Hz, 1H), 7.67 (dd, *J* = 7.5, 1.4 Hz, 1H), 6.98 (t, *J* = 1.5 Hz, 1H), 6.82 (dd, *J* = 4.4, 3.0 Hz, 3H). ^13^C NMR (150 MHz, DMSO-d_6_) δ: 160.74, 157.10, 151.07, 145.40, 136.07, 134.17, 129.72, 127.23, 124.27, 117.41, 117.30, 115.27, 107.65. Anal. Calcd for the C_15_H_8_BrCl_2_NO: C, 48.82; H, 2.18; N, 3.80; O, 4.34. Found: C, 48.75; H, 2.22; N, 3.84; O, 4.39.

6-bromo-4-(3-chloro-4-fluorophenoxy)quinoline (**3f**). M.p.: 300–302 °C. IR (KBr, cm^−1^): 2925, 2869, 1670, 1654, 1595, 1507, 1287, 1147, 1090, 922, 830. ^1^H NMR (600 MHz, DMSO-d_6_) δ: 8.45 (d, *J* = 7.5 Hz, 1H), 8.13 (d, *J* = 1.4 Hz, 1H), 8.05 (d, *J* = 7.5 Hz, 1H), 7.67 (dd, *J* = 7.5, 1.4 Hz, 1H), 6.92 (dd, *J* = 5.0, 1.4 Hz, 1H), 6.82 (t, *J* = 7.2 Hz, 2H), 6.70 (ddd, *J* = 7.4, 5.0, 1.5 Hz, 1H). ^13^C NMR (150 MHz, DMSO-d_6_) δ: 160.74, 154.22, 152.29, 152.26, 152.12, 151.07, 145.40, 134.17, 129.72, 127.23, 122.93, 122.72, 121.54, 121.49, 117.82, 117.76, 117.41, 117.08, 116.87, 115.27, 107.65. Anal. Calcd for the C_15_H_8_BrClFNO: C, 51.10; H, 2.29; N, 3.97; O, 4.54. Found: C, 51.05; H, 2.33; N, 4.01; O, 4.56.

6-bromo-4-(4-(methylthio)phenoxy)quinoline (**3g**). M.p.: 305–306 °C. IR (KBr, cm^−1^): 3095, 2954, 2923, 2853, 1458, 1261, 1235, 1065, 874, 811. ^1^H NMR (600 MHz, DMSO-d_6_) δ: 8.47 (d, *J* = 7.5 Hz, 1H), 8.13 (d, *J* = 1.4 Hz, 1H), 8.05 (d, *J* = 7.5 Hz, 1H), 7.67 (dd, *J* = 7.5, 1.6 Hz, 1H), 7.06 (d, *J* = 7.5 Hz, 2H), 6.86 (d, *J* = 7.5 Hz, 1H), 6.74 (d, *J* = 7.5 Hz, 2H), 2.63 (s, 3H). ^13^C NMR (150 MHz, DMSO-d_6_) δ: 160.74, 154.32, 151.07, 145.40, 134.93, 134.17, 129.72, 129.49, 127.23, 121.00, 117.41, 115.27, 107.65, 16.53. Anal. Calcd for the C_16_H_12_BrNOS: C, 55.50; H, 3.49; N, 4.05; O, 4.62. Found: C, 55.47; H, 3.52; N, 4.09; O, 4.66.

Methyl 4-(6-bromoquinolin-4-yloxy)benzoate (**3h**). M.p.: 324–326 °C. IR (KBr, cm^−1^): 3095, 2954, 2923, 2853, 1458, 1261, 1235, 1065, 874, 811. ^1^H NMR (600 MHz, DMSO-d_6_) δ: 8.47 (d, *J* = 7.5 Hz, 1H), 8.14 (d, *J* = 1.4 Hz, 1H), 8.05 (d, *J* = 7.5 Hz, 1H), 7.81 (d, *J* = 7.5 Hz, 2H), 7.67 (dd, *J* = 7.5, 1.6 Hz, 1H), 6.96 (d, *J* = 7.5 Hz, 2H), 6.85 (d, *J* = 7.5 Hz, 1H), 3.95 (s, 3H). ^13^C NMR (150 MHz, DMSO-d_6_) δ: 167.35, 160.74, 158.93, 151.07, 145.40, 134.17, 131.14, 129.72, 127.23, 124.51, 119.26, 117.41, 115.27, 107.65, 52.08. Anal. Calcd for the C_17_H_12_BrNO_3_: C, 57.00; H, 3.38; N, 3.91; O, 13.40. Found: C, 56.92; H, 3.42; N, 3.96; O, 13.41.

### 3.4. Antibacterial Activity

#### 3.4.1. Agar Well Diffusion Method

Agar well diffusion assay against ESBL producing *E. coli*. was employed to study the antibacterial effect. First, 0.5 McFarland bacterial suspension was injected over the Mueller Hinton Agar plate and a hygienic 6 mm cork borer was employed to build wells on each plate. Then, 100 μL from every DMSO dilution (50 mg, 40 mg, 30 mg, 20 mg, 10 mg) was poured into every well, and plates were nurtured at 37 °C overnight. A Vernier caliper was used to dignify the region of inhibition (mm). The results were obtained thrice to minimize the chance of error. Meropenem (10 µg) disc was utilized as antibiotic control [43].

#### 3.4.2. Identification of the Bacterial Strains

*Escherichia coli* and *S. aureus* were clinically isolated from blood samples using BACTEC/Alert (BD, UK). These isolates were developed on plasma and MacConkey agar (Oxoid, UK) and confirmed by VITEK 2 compact system (Biomerieux, France).

#### 3.4.3. Antibiogram of the Isolates

The MIC (µg/mL) of numerous antiseptics against these pathogens was carried out in VITEK 2^®^ compacted procedure (BioMerieux, France). The antimicrobials compounds verified were penicillin, cefoxitin, ampicillin/sulbactam, ticarcillin/clavulanic acid, piperacillin, cefuroxime, cefixime, ceftriaxone, cefepime, aztreonam, meropenem, levofloxacin, moxifloxacin, minocycline, tetracycline, tigecycline, chloramphenicol, colistin, and trimethoprim. The interpretation of the susceptibility was done as per CLSI guidelines 2020 (Table 9).

#### 3.4.4. Phenotypic Determination of MRSA

MRSA was calculated by the agar disc diffusion method as defined by the CLSI 2020. In short, MRSA was poured into the Mueller Hinton agar (MHA) plate and a cefoxitin (30 µg) disc was placed. If the zone of inhibition was ≤21 mm, it was considered MRSA positive [44].

Gram-negative bacterial strains were resistant to the AWaRe (Access, Watch, and Reserve) WHO classes of antibiotics such as β-lactams, aminoglycosides, and quinoline except for polymyxin. Further, MRSA displayed resistance to β-lactams, and quinoline and was sensitive to vancomycin and linezolid.

#### 3.4.5. Phenotypic Detection ESBL Enzyme

The Gram-negative isolates were verified for the ESBL activity using the double-disc diffusion method as described previously [45]. Briefly, the isolates were loaned on the MHA plate, co-amoxiclav antibiotic was placed in the center, and 3rd generation cephalosporins were placed at the distance of 10 mm to co-amoxiclav as shown in the Figure 9. After overnight incubation, a synergy-like pattern confirmed the presence of ESBL enzymes.

*E. coli* were ESBL producers and *Staphylococcus aureus* were MRSA.

#### 3.4.6. Minimum Inhibitory Concertation of Different Compounds against XDR Pathogens

The MIC (% *w/v*) of different derivatives was investigated by employing micro broth weakening assay. The 2–3 isolated colonies were assorted in 20 mL of double-potency lysogeny potage (LB) intermediate in 50 mL of falcon tube and protected at 37 °C instantaneously. The dilutions of the bacterial suspensions were done to attain 0.5 McFarland at an optical density (OD) of 0.07 at 600 nm. Briefly, sequential reductions in the concentration of each compound (0.76, 1.56, 3.12, 6.25, 12.5, 25, 50 mg) were prepared in DMSO and 100 μL of each compound diluted solution was added in 96 wells, flat-bottom microliter plates (Thermo Fisher Scientific, Leicestershire, UK). Then, 100 μL of bacterial suspension was poured separately into all the wells. Negative-control wells had 100 μL of LB while positive-resistor wells had LB by microbial deferment. The microtiter plate was protected at 37 °C instantaneously in a pulsating incubator (MaxQTM Mini 4450, Thermo Fisher Scientific) at 3 g. MIC was deliberated by associating every well with high-value and low-value regulator shafts [46]. All trials were executed thrice.

#### 3.4.7. Minimum Bactericidal Concentration Determination

The first diluted solution with zero progression on an agar plate is named the minimum bactericidal concentration (MBC, % *w/v*). A concentration of 10 μL from the zero-evolution wells of a microtiter plate was added to the source of nourishing gelatinous carbohydrate plates (Oxoid, Hampshire, UK) and nurtured at 37 °C for 24 h in the oxygenated environment. Cell sustainability and any cell growing were examined by examining the plates and they were categorized as bacterial progression and zero or no microbial growth [43]. All trials were performed thrice.

#### 3.4.8. Molecular Docking Study

Molecular docking studies were accomplished to identify the stable interactions of synthesized compounds (**3b**, **3c**, **3d**, **3e**, **3h**) against *E. coli* curli fiber biogenesis biofilm receptor protein using Auto Dock Tools version 1.5.6 (ADT) software [47]. The dataset of compounds was categorized into three activity levels: highly active (**3e**), moderately active (**3c**, **3h**), and least active (**3b**, **3d**). Moreover, the X-ray crystallographic structure of *Escherichia coli* curli fiber biogenesis biofilm receptor protein (PDB ID: 2Y2T) was extracted from the Protein Data Bank (PDB) at a resolution of 2.30 Å [18]. The structure was prepared before the docking by the removal of water and protonation of the 3D structure at pH 7.4 using Molecular Operating Environment (MOE) version 2019.0237 [19]. Moreover, the energy minimization of 2Y2T was performed using the Amber99 force field in MOE [20]. The structure of synthesized compounds (active, moderately active, and least active) was optimized using Merck Molecular Force Field (MMFF94) [48] basis set in the MOE. The grid box dimensions were set in autodock vina to cover the entire receptor through the auto grid. Furthermore, both the protein (2Y2T) and compounds (**3a**–**3h**) were considered flexible by performing a total of 10 genetic algorithm runs per molecule to enhance the conformational space and to explore their interactions within the binding cavity of protein. Furthermore, to elaborate the ligand and receptor binding energy/affinity by the search parameter Genetic Algorithm, the ADT software package was used.

### 3.5. Computational Studies

The density functional theory (DFT) study was performed to predict the electronic distribution, electrostatic, and structural properties [49] of the newly synthesized products, by Gaussian 09 software [50], and CYL view and Gauss View 05 software [51] were employed to study their structural visualizations by the analysis of frontier orbitals, molecular electrostatic potentials, and reactivity parameters

## 4. Conclusions

In this study, the proficient synthesis of the novel derivatives (**3a**–**3h**) is demonstrated by reacting different aryl boronic acids with 6-bromoquinolin-4-ol (**1a**) by Chan–Lam coupling reactions in modest to good yields. Both electron-donating as well as withdrawing groups present on the boronic acids interacted with the quinoline moiety. All these new derivatives were confirmed by IR, ^1^H NMR, and ^13^C NMR techniques. All these molecules exhibited their relative peaks of protons and carbons at their specific ppm zone. In case of doublet, triplet, or doublet of doublet peaks, their *J* values were also measured in Hz to confirm their neighboring protons. DFT studies proved that **3a**, **3e**, **3f**, and **3h** are more stable and less reactive, while **3b**, **3c**, **3d**, and **3g**) are more reactive and less stable. The synthesized compounds **3a**–**3h** exhibited remarkable antibacterial activities against ESBL producing *E.coli* and *Staphylococcus aureus* (MRSA) by agar well diffusion method. MICs and MBCs were measured by a micro broth dilution test. All the compounds showed bacterial inhibition activity at varied concentrations but **3d** unveiled improved antiseptic characteristics with MIC values of 25 mg and 12.5 mg and MBC values of 50 mg and 25 mg against ESBL producing *E. coli* and MRSA. Molecular docking studies were also accomplished to document the interaction of the synthesized compounds **3a**–**3h** against *E. coli* curli fiber biogenesis biofilm receptor protein using Auto Dock Tools version 1.5.6 (ADT) software. It was found from a molecular docking study that compound **3e** is highly unstable and highly active, **3c** and **3h** moderately active, while **3b** and **3d** are more stable and least active. Hence, in vitro activity and molecular docking revealed the antiseptic perspective of **3d** and these impending antibacterial quinoline derivatives might be utilized in a dimension to synthesize new antiseptic drugs in near future.

## Data Availability

Data are contained within the article and the Appendix A.

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
