# Peer review of "Facile Synthesis of Functionalized Phenoxy Quinolines: Antibacterial Activities against ESBL Producing Escherichia coli and MRSA, Docking Studies, and Structural Features Determination through Computational Approach"

_molecules, 2022, doi:10.3390/molecules27123732_

Round 1

Reviewer 1 Report

Abstract:

Results and conclusions are missed from the abstract.

introduction:

  • Extended spectrum-β-lactamase producing Escherichia coli (ESBL E. coli) and methicillin-resistant Staphylococcus aureus (MRSA) produce plenty of antibiotics: what is meant? is it true?
  • Line 86: to analyze about the antibacterial activities of the: there are errors in writing.
  • aims of the study should be written at the end of introduction.
  • names of bacteria should be written in italicMethods:
  • general information 
    • write the source of materials used.

Author Response

Answers to the reviewer

Firstly, thanks so much for your time and your helpful specific comments on our manuscript and I have revised the manuscript, accordingly, taking your comments in considerations.

Reviewer 2 Report

The manuscript entitled “Facile Synthesis of functionalized Phenoxy quinolones: Antibactial activities against clinically isolated ESBL producing Eschercia coli and methicillin resistant Staphylococcus aureus; their docking studies and structural features determination through Computational approach” by Mahwish Nadeem et al. described the synthesis of 4-hydroxy quinoline derivatives by the reaction of 6-bromoquinolin-4-ol with 4-methyl phenylboronic acid in the presence of Et3N and Cu(OAC)2, and study obtained compounds as potential  antimicrobial agents. The manuscript may be of general interest to the researchers of this field, but the manuscript lacks some information that the author should consider and incorporate in the present form of the manuscript. Here are a few concerns that need to be addressed in the present form of the manuscript.

  1. The title of the manuscript should be shortened! and all words should be in capital letters.
  2. The abstract should be corrected in accordance with the rules of journal for abstract structure, namely the abstract should contain background addressed in a broad context and highlight the purpose of the study; the method for antibacterial study should be added and the main conclusions should be indicated in the abstract.
  3. “6-Bromoquinolin-4-ol” and “4-methyl phenylboronic acid” should be added instead of “Hydroxy quinolone”
  4. “Chan-lan”, “chan-lam” should be “Chan–Lam”
  5. “Et3-N”, “Et3-N” should be “Et3N” or “TEA”
  6. It should be added an information of commercial suppliers and equipment for IR and elemental analysis in “3.1. General Information”.
  7. The chemical names of the obtained compounds in the experimental part should be in capital letters. The author should carefully check the names of obtained compounds: there are quinolines, not “quinolones”.

Author Response

Answers to the reviewer

Firstly, thanks so much for your time and your helpful specific comments on our manuscript and I have revised the manuscript, accordingly, taking your comments in considerations.

Reviewer(s)' Comments to Author:

  1. The title of the manuscript should be shortened! and all words should be in capital letters.

The title of the manuscript has been revised in the main manuscript.

  1. The abstract should be corrected in accordance with the rules of journal for abstract structure, namely the abstract should contain background addressed in a broad context and highlight the purpose of the study; the method for antibacterial study should be added and the main conclusions should be indicated in the abstract.

The abstract has been modified according to the reviewer comments in the main manuscript.

  1. “6-Bromoquinolin-4-ol” and “4-methyl phenylboronic acid” should be added instead of “Hydroxy quinolone”

The changes have been incorporated in the main manuscript.

  1. “Chan-lan”, “chan-lam” should be “Chan–Lam”

The changes have been incorporated in the main manuscript.

  1. “Et3-N”, “Et3-N” should be “Et3N” or “TEA”

The changes have been incorporated in the main manuscript.

  1. It should be added an information of commercial suppliers and equipment for IR and elemental analysis in “3.1. General Information”.

The changes have been incorporated in the main manuscript.

  1. The chemical names of the obtained compounds in the experimental part should be in capital letters. The author should carefully check the names of obtained compounds: there are quinolines, not “quinolones”

The changes have been incorporated in the main manuscript.

Reviewer 3 Report

The presented manuscript describes the development and research of new antimicrobials. This issue is current and useful. The work brings new, active and usable structures. I appreciate that the article combines synthetic work, in vitro testing and computational studies.

The choice of receptor for docking would be explained in the clarity of the article. I also miss a description and rationale of the design of the novel structures.

Author Response

Answers to the reviewer

Firstly, thanks so much for your time and your helpful specific comments on our manuscript and I have revised the manuscript, accordingly, taking your comments in considerations.

Reviewer(s)' Comments to Author:

Q: The choice of receptor for docking would be explained in the clarity of the article. I also miss a description and rationale of the design of the novel structures.

The choice of receptor for docking has been explained in the main manuscript.
